# A synthesis dataset of permafrost thermal state for the Qinghai-Xizang (Tibet) Plateau, China

Lin Zhao[1,2*], Defu Zou[2], Guojie Hu[2*], Tonghua Wu[2], Erji Du[2], Guangyue Liu[2], Yao Xiao[2], Ren Li[2], Qiangqiang Pang[2], Yongping Qiao[2], Xiaodong Wu[2], Zhe Sun[2], Zanpin Xing[2], Yu Sheng[3], Yonghua Zhao[2], Jianzong Shi[2], Changwei Xie[2], Lingxiao Wang[1], Chong Wang[1], Guodong Cheng[2]

[1] *School of Geographical Sciences, Nanjing University of Information Science & Technology, Nanjing 210044, China*

[2] *Cryosphere Research Station on Qinghai-Xizang Plateau, State Key Laboratory of Cryospheric Sciences, Northwest Institute of Eco-Environment and Resources, Chinese Academy of Sciences, Lanzhou 730000, China*

[3] *State Key Laboratory of Frozen Soil Engineering, Northwest Institute of Eco-Environment and Resources, Chinese Academy of Sciences, Lanzhou 730000, China*

**Correspondence:** Lin Zhao (lzhao@nuist.edu.cn), Guojie Hu (huguojie123@lzb.ac.cn)

**Abstract:**

Permafrost is important for the climatic, hydrological, and ecological processes on the Qinghai-Xizang (Tibet) Plateau (QXP). The changing permafrost and its impact have been attracting great attention worldwide never before, and more observational and modeling approaches are need to promote an understanding of permafrost thermal state and climatic conditions on the QXP. However, there were almost no synthesis dataset on the permafrost thermal state and climate background on the QXP, but were sporadically reported in different literatures due to the difficulties to access to and work in this region, where the weather is severe, and environmental conditions are harsh and the topographic and morphological features are complex. In this study, a comprehensive dataset under quality controlled consisting of long-term meteorological, ground temperature, soil moisture and soil temperature data were compiled from an integrated, distributed and multiscale observation network in the permafrost regions of the Cryosphere Research Station on the QXP. Meteorological data were observation by automatic meteorological stations from a comprehensive observation network. The soil temperature and moisture data were collected from an integrated observation



system in the active layer. Deep ground temperature was observed from boreholes. These datasets
were helpful for the scientists with multiple study fields (i.e., climate, cryospheric, ecology and
hydrology, meteorology science), which will greatly promote the verification, development and
improvement of the hydrological model, land surface process model and climate model on the QXP.
The datasets are available from the National Tibetan Plateau/Third Pole Environment Data Center
(https://data.tpdc.ac.cn/en/disallow/789e838e-16ac-4539-bb7e-906217305a1d/,        doi: 10.11888/
Geocry.tpdc.271107).

**1 Introduction**
Permafrost is widely distributed on the QXP, which is called the "Third Pole of the Earth" (Qiu,
2008), is about $1.06 \times 10^6 \mathrm{km}^2$ in area and accounting for approximately a quarter of the QXP (Zou
et al., 2017). Its unique and complicated hydrothermal process has great regulating effects on ground
surface moisture, energy and mass exchange, ecosystem stability and carbon cycles (Cheng et al.,
2019; Schuur et al., 2011). The surface energy and water cycle over the QXP have great influence
on Asian monsoon, East Asian atmospheric circulation and global climate change (Ma et al., 2017;
Yao et al., 2017). The characteristics of diabatic heating field of QXP are also used as an important
factor for the short-term climate prediction in China (Liu and Hou, 1998; Wu et al., 2009; Ye and
Gao, 1979).
The permafrost in the QXP has experienced significant degradation in response to climate
warming, which mainly manifested as the permafrost area shrinking and ground temperature rise,
the increased active layer, and decreased permafrost thickness (Hu et al., 2019b; Sharkhuu et al.,
2007; Wang et al., 2000; Cheng et al., 2019). The permafrost degradation has caused the changes
of surface vegetation characteristics. It was reported that the area of Alpine meadow on the QXP
decreased by $16.2 \times 10^4 \mathrm{km}^2$ (accounted for 32.4% of the QXP (Zhao and Sheng, 2015)) in recent
decades, which caused the change in hydrological processes, ecological environment and further
led to desertification (Cheng and Jin, 2013; Cheng et al., 2019; Wu et al., 2003; Zhao et al., 2019).
In addition, permafrost degradation could result in the decomposition of organic matter and
greenhouse gases increased, which will finally affect the surface energy balance and the climate
system (Wang et al., 2006a; Ping et al., 2015; Schuur et al., 2015; Schuur et al., 2011; Wu et al.,
2012; Hu et al., 2019a). Permafrost degradation have also altered the geomorphological features
and affected the stability of engineering structures in this region (Zhao et al., 2017).



However, the collection of long-term and high resolution data over the permafrost regions of
QXP is challenging due to the complex terrain, severe weather, and inconvenient to access (Ma et
al., 2008; Li et al., 2012). Previous studies on the permafrost are focused on local and sites scale
and major along the Qinghai-Tibet highway/Railway (Cuo et al., 2015; Su et al., 2013). Some new
observation sites in permafrost regions in the vast western territory of the QXP were reported in
recent years (Zhao et al., 2017; Zhao et al., 2018; Zhao et al., 2020). It is urgent to establish a
synthesis observational database of permafrost thermal state and its climatic background to satisfy
the requirements of calibration and validation for remote sensing interpretation and hydrothermal
processes simulation, and also for the key parameters acquisition in permafrost regions (Bao et al.,
2016; Li and Koike, 2003; Wang et al., 2017; Zhang et al., 2008; Hu et al., 2020). The complexity
of the dynamic process of water and heat in freeze-thaw cycles is also considered to be one of the
important reasons why the land surface model is not effective in simulating the permafrost change
(Chen et al., 2014; Hu et al., 2016; Yang et al., 2018). Nevertheless, it is of great significance to
provide a set data in thermal dynamic characteristics of the permafrost on the QXP (Wang et al.,
2006b; Zhao et al., 2004).
The Cryosphere Research Station on the Qinghai–Xizang Plateau, Chinese Academy of
Sciences (CRS-CAS), has established a comprehensive and widely permafrost monitoring network
on the QXP (Zhao et al., 2019, 2020). This network is mainly focus on monitoring permafrost and
its environmental factors in very high and cold regions of the QXP. Since the station was established
in 1987, we have conducted long-term continuous monitoring and large-scale field investigations
on permafrost of the QXP, and thus synthetically studied the mechanisms of the change in
hydrothermal conditions of permafrost and their simulations and ecological effects. This paper first
integrated air temperature, ground temperatures, soil moisture and permafrost temperature dataset
over the permafrost regions across QXP from the CRS-CAS monitoring networks. The
comprehensive permafrost monitoring network is summarized in Sect. 2. In Sect. 3, the datasets are
described in detailed, with data evaluated and analysis. In Sect. 4, the data availability and access
are provided, and in Sect. 5, the conclusions and future work are summarized.
**2 Monitoring networks and data processing**
**2.1 Permafrost monitoring networks**
The thermal state of permafrost is mainly controlled by climate, and affected by land surface
and geological conditions. The ecosystem in the permafrost region mainly dominated by alpine
meadow, swamp meadow, alpine steppe, and alpine desert (Wang et al., 2016). The soils in the
western permafrost region are Gelisols, Inceptisols and Aridisols, and in the eastern mainly consists
of Gelisols, Mollisols and Inceptisols (Li et al., 2015). The permafrost monitoring network include
6 automatic meteorological stations, 8 active layer sites and, 77 boreholes (Fig. 1, Table 1). The
elevation of all the sites are extremely high, exceeding 4000 m above the sea level (31.82~37.75 °N,
77.58~99.50 °E).

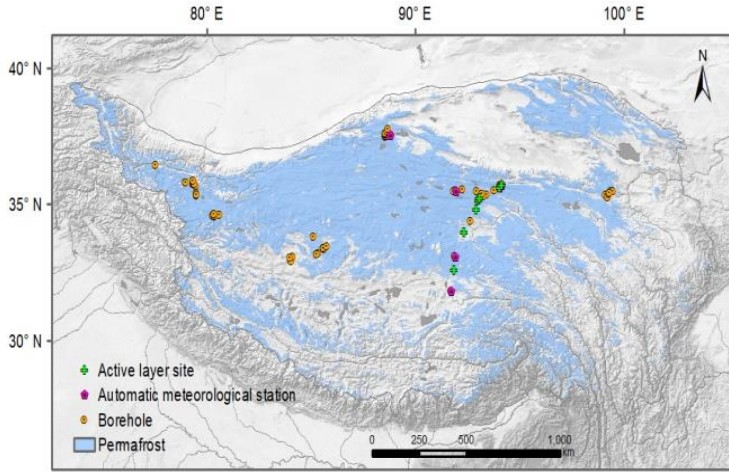


**Figure 1.** The permafrost monitoring networks on the QXP
There are only 4 national meteorological stations in the vast territory of permafrost regions.
We set 6 automatic meteorological stations (Figure 2) within the permafrost zone gradually since
2004, the main observation indices include air temperature, humidity, wind speed gradient
observation, radiation balance, and precipitation, etc. These six stations also include an active layer
observation system and borehole monitoring simultaneously record frozen soil temperature, and are
represent different permafrost, climate, vegetation, soil and other characteristics in different regions
of the QXP. LDH has the lowest latitude, and it gets the warmest air temperature and the largest
annual precipitation, while TSH and AYK which located in the northwest and north of the plateau,
respectively, have the minimum and penultimate temperatures and annual precipitations. TSH has
the highest solar radiation among the 6 stations.





XDT and TGL are two sites with the longest sequence of 6 gradient meteorological stations.
They were established in May 2004 and have had data accumulation over 16 years. XDT is located
in the northern part of the QXP, near the northern permafrost boundary of the QXP. It represents
the characteristics of the island-shaped permafrost area. TGL site is located on the north side of the
Tanggula Mountains in the hinterland of the QXP and represents the characteristics of the
continuous permafrost area. LDH is a new site established in 2014, is also located along the Qinghai-
Tibet Highway. It is near the southern boundary of the permafrost region on the QXP, and can
represent the characteristics of the discontinuous permafrost region. ZNH is located in the source of
the Yangtze River, is an extremely rare observation station established in the vast unmanned area
of the Qiangtang Regions of the QXP. It fills the data gap in the central and northern areas of the
QXP and also located in a continuous permafrost area. AYK is located in the Altun Mountains area
in the northern Tibetan Plateau, which is also a vast unmanned area as Qiangtang Region and is one
of the areas with few observations. TSH is located in the West Kunlun Mountain area where is also
an inaccessible area, is at the western border of the permafrost region on the QXP. It can reflect the
regional characteristics of arid, cold, and high altitude in the western part of the QXP. The ground
temperature and soil moisture observed of active layer and permafrost was summarized in Table 1.

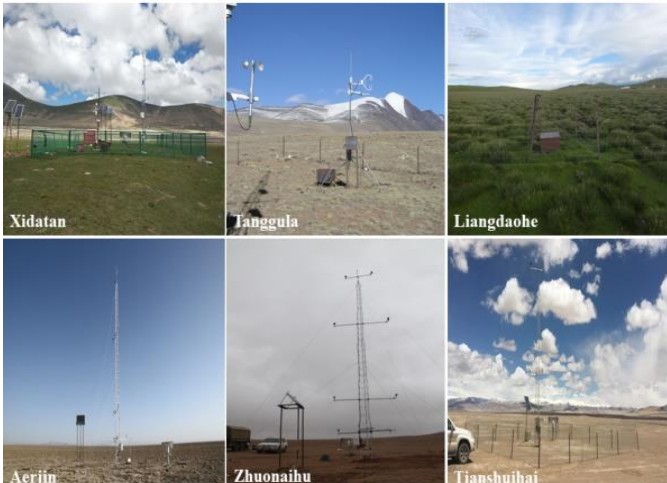


**Figure 2.** The six comprehensive meteorological stations

**Table 1** The observation instruments and items for meteorological data, ground temperature and soil water content

| Observation site type | Available sites | Observation item | Instrument | Accuracy | Height/Depth |
|---|---|---|---|---|---|





| | | Upward/downward short-wave radiation | CM3, Kipp & Zonen, Holland | ±10% | 2 m |
|---|---|---|---|---|---|
| **Meteorological Stations** | 6 | Upward/downward long-wave radiation | CM3, Kipp & Zonen, Holland | ±10% | 2 m |
| | | Air temperature | HMP45C, Vaisala Finland | ±0.5 ℃ | 2,5,10 m |
| | | Air humidity | | ±3% RH | 2,5,10 m |
| | | Wind velocity | 05103_L/RM, Campbell, USA | ±0.3 m/s | 2,5,10 m |
| | | Precipitation | T-200B Precipitation Gauge | ±0.1 mm | 5 m away |
| **Active Layer** | 10 | Soil temperature | 105T/109 Thermocouple temperature sensor | ±0.1 ℃ ±0.2 ℃ | 0.5 m,1.0 m,2 m, >2 m |
| | | Soil moisture content | CS616/ Hydra Soil moisture sensor | ±2.5% | |
| **Borehole** | 40 | Ground Temperature | Thermistor, SKLFSE, CHINA | ±0.05 ℃ | 0.4 m - 34.5 m (total 47 layers) |

## 2 Monitoring data

The main observation items and instruments for the meteorological observations were shown in Table 1. The observation was done every 10 minutes, and was averaged and recorded every 30 minutes automatically. The data were recorded by CR10X, CR1000 and CR3000 data logger (Campbell Scientific). Meteorological data (e.g., the precipitation, radiation, air temperature, relative humidity and wind speed) were recorded hourly with a CR1000/CR3000 data acquisition instrument (Campbell Scientific Inc., USA) (Fig 3a). There were three measured at heights of 2 m, 5 m and 10 m for air temperature, relative humidity and wind speed (Table 3).

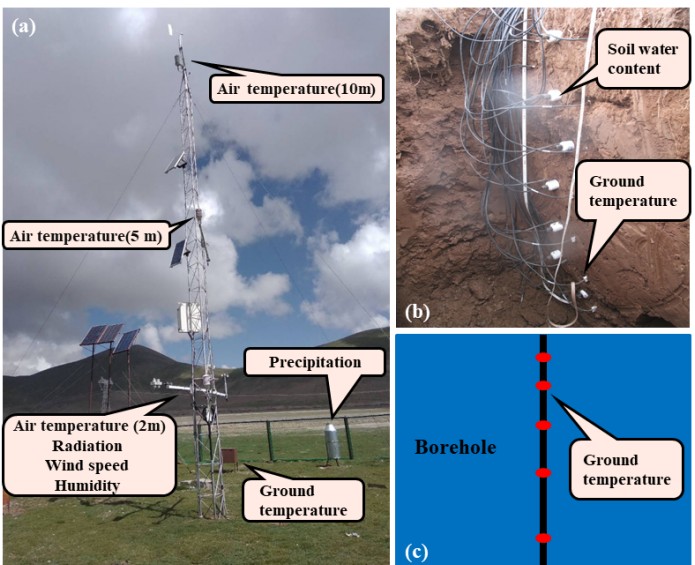


**Figure 3.** The comprehensive observation system : (a) meteorological observation, (b) ground temperature and soil

water content in the active layer and (c) ground temperature observation for permafrost.

The ground temperature was measured at different depths from ground surface to the depth of
10 to 50 cm below the permafrost table with a 105T/109 thermocouple Probe with an accuracy of $\pm$
0.1 ℃/$\pm$0.2 ℃ in the active layer (Fig 3b). The soil water content was measured by a Hydra soil
moisture sensor (Table 1), by connecting to a CR10X/CR1000/CR3000 data logger (Campbell
Company, USA).
The ground temperature in the borehole was measured by the Thermistors (with an accuracy
of $\pm$0.1 ℃) produced by the State Key Laboratory of Frozen Soil Engineering, Cold and Arid
Regions Environmental and Engineering Research Institute of the Chinese Academy of Sciences
(SKLFSE, CAREERI, CAS), which were installed in the boreholes and connected to an automatic
data logger (CR1000/ CR3000, Campbell Scientific Company, Logan, UT, USA) to monitor ground
temperatures at various deep depths (Table 1) (Fig 3c).
**2.3 Data processing workflow**
Data processing workflow is showed in Figure 4. All data are under quality control before
processing. The quality control was two-fold: (1) the missing data were replaced by -6999; (2) the
singular unphysical data were rejected, and the gaps were replaced by -6999. In addition, all the





daily data were calculated by every 30 min (1 h) interval per day. The active layer thickness was
derived by the maximum depth of 0 ℃ isotherm from linear interpolation of the daily maximum
ground temperature. The monthly means air and ground temperatures, radiation, wind speed,
relative humidity and soil water content were also calculated. The dataset also provides mean-annual
air temperature (MAAT); mean-annual ground temperature at depth of 0.5 m, 0.75 m, 1.0 m, 1.5 m,
2 m and >2 m; and maximum and minimum ground temperature. The trend of air temperature, active
layer thickness, and ground temperature is analyzed and provided at the stations with long time
observation.

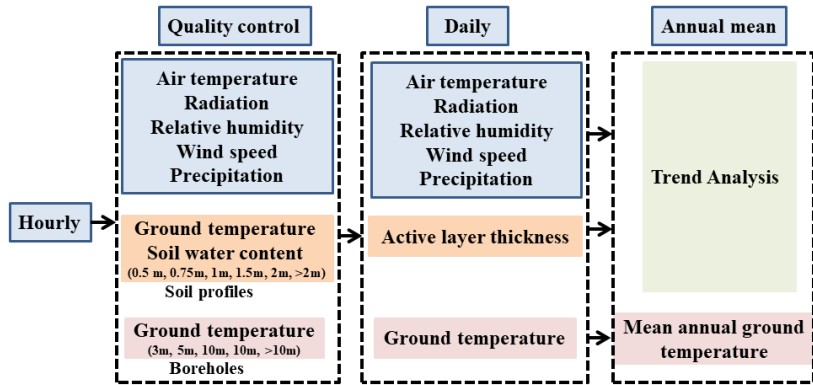


**Figure 4**. Schematic diagram of data processing workflow used to compile the permafrost dataset on the QXP.
**3 Data description and evaluation**
**3.1 Meteorological data**

**Table 2.** The information of six meteorological stations

| Sites | XDT | TGL | LDH | ZNH | AYK | TSH |
|---|---|---|---|---|---|---|
| Latitude (ºN) | 35.72 | 33.07 | 31.82 | 35.49 | 37.54 | 35.62 |
| Longitude (ºE) | 94.13 | 91.94 | 91.74 | 91.96 | 88.8 | 94.06 |
| Elevation (masl) | 4538 | 5100 | 4808 | 4784 | 4300 | 4844 |
| Vegetation | Alpine steppe | Alpine meadow | Alpine wet meadow | Alpine desert | Alpine desert | Alpine desert steppe |




| Observation height above the ground surface (m) | 2,5,10 | 2,5,10 | 2,5,8 | 2,4,10,15 | 2,4,10,15 | 2,4,10 |
|---|---|---|---|---|---|---|
| Data since | 2004.5 | 2004.5 | 2014.9 | 2013 | 2013 | 2015 |
| Missing date | 17.7~17.9 | 09.6~09.8, 16.10~17.9, 17.12~18.1, | 18.1~18.4 | 14.7~15.4 (Ta, Preci.) | 15.8~15.9 | Ws at 2m |
| Ta (ºC) | -3.6 | -4.7 | -2.3 | -4.9 | -5.2 | -6.0 |
| RH (%) | 53.5 | 51.5 | 48.2 | 53.9 | 46.1 | 40.6 |
| Precipitation (mm) | 384.5 | 352.0 | 388.6 | 277.8 | 158.6 | 103.3 |
| Wind speed (m/s) | 4.1 | 4.1 | 3.2 | 4.7 | 4.5 | |
| DSR (W/m$^2$) | 224.2 | 233.4 | 231.4 | 204.8 | 198.2 | 250.8 |
| USR (W/m$^2$) | 66.8 | 61.4 | 46.6 | 46.3 | 53.8 | 68.5 |
| DLR (W/m$^2$) | 223.0 | 214.8 | 237.2 | 233.8 | 223.0 | 211.5 |
| ULR (W/m$^2$) | 304.5 | 304.5 | 315.9 | 303.2 | 307.6 | 311.3 |
| Net radiation | 75.9 | 82.3 | 106.0 | 89.2 | 59.8 | 82.5 |

The mean annual air temperatures (MAAT) of all 6 sites are between -2.3 ~ -6 ºC, and their
seasonal variation are significant (Fig. 5). The mean monthly air temperatures in 3 seasons except
summer are lower than 0 ºC. The difference of the air temperatures between these stations were
mainly caused by the difference in altitude and latitude. Although the difference in summer is
relatively small, it is very large in winter, and is obvious in spring and autumn.
Precipitation shows significant seasonal variation, which is closely related to the monsoon
cycles. The precipitation from May to September is more than 85% of its annual amounts at the
sites other than TSH (78.6%). Most of the precipitation is concentrated in summer. A small amount
is in late spring and early autumn, and rare in the winter. Precipitation has significant spatial
difference, which is more than 350 mm in average at XDT, TGL, LDH along the Qinghai-Xizang
Highway, and is relatively higher than other sites in the western QXP. The precipitation at ZNH,
located in the hinterland of the plateau, is slightly lower, and is about 150 mm (slightly higher than

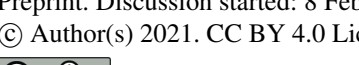



half at ZNH) in AYK and much lower, which is located on the northern edge of the Tibetan Plateau
and has the highest latitude among all the 6 sites. The annual total precipitation at TSH is the lowest
of all the observation sites, and is only 100 mm, which is located near the western boundary of QXP.

The air humidity exhibits seasonal variation, which are very consistent with the change of air

temperature and precipitation. The difference between the stations is related to the precipitation,
especially in summer. Due to the scarce precipitation, the relative humidity at TSH is at a low level
throughout the year. And it is similar in AYK. It is worth noting that the relative humidity in TGL
and LDH is quite low in winter. Since their latitude are lower, the air temperatures in winter are
higher. The ground evaporates more and becomes drier. The wind speeds at all stations on the
plateau are generally high. Except LDH, the average annual wind speeds are all above 4 m/s. The
wind speed is the highest in winter, followed by spring and the lowest in summer. The wind speed
of LDH shows a low state throughout the year, due to its latitude and local conditions. It is located
in a well-developed swamp meadow area with a gentle slope, where the vegetation is lush.

Downward shortwave radiation (total solar radiation) usually reaches its maximum in May, at

most sites except TSH. This is due to the concentration of rain in summer. Therefore, the mean
downward shortwave radiation in summer is only slightly higher than that in spring. However, at
TSH (with little precipitation), it is very high in summer, and also significantly higher than other
sites in spring and autumn. The upward shortwave radiation is mainly restricted by the surface
albedo. Its high value is mainly affected by the snow on the ground. Its seasonal changes reflect that
the snow cover may mainly appear in autumn, followed by spring and relatively little in winter. The
upward short-wave radiation of TSH in all seasons is high, which is related to the area with less
precipitation and the surface is almost dry and bare. The upward and downward long-wave radiation
is closely related to air temperature and surface temperature, respectively, and their seasonal
variation trend is basically consistent with the change of air temperature.

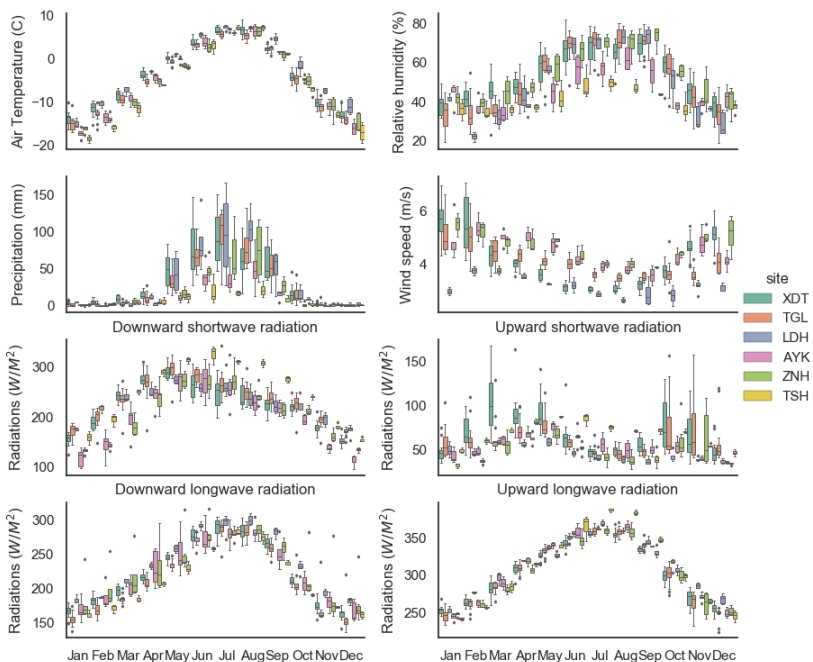


**Figure 5.** Characteristics of monthly observation variables at six meteorological stations

XDT and TGL stations can provide basic data for physical process research and land surface

process model research. The annual mean temperature of the two stations showed increasing trends,

with rates of 0.66 and 0.40 ºC/10a, and p-values of 0.27 and 0.23, respectively. The warming trend

is the highest in summer and autumn, and has a good significance. However, the temperature in

winter shows a weak decrease. The precipitations show an insignificant week decrease trend (-15.0

and -14.3 mm/10a). It shows a slightly decreasing trend in summer and autumn, and an increasing

trend in spring (Fig. 6).

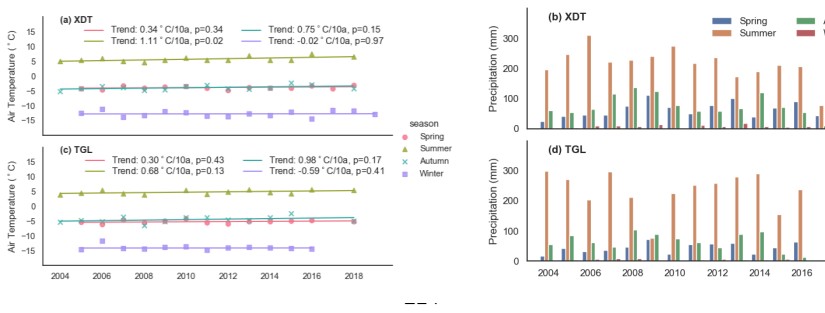





**Figure 6.** Seasonal mean series and changes of temperature and precipitation at XDT and TGL from 2004 to 2018

**3.2 Active layer thickness**

The active layer thicknesses varied from about 120 cm to about 300 cm along Qinghai-Tibet highway under different surface vegetation conditions (Fig.7). Ch04, which locates at sporadic island permafrost of the QTB southern permafrost distribution limit regions under swamp meadow condition, appeared as the shallowest active layer site. Its average thickness was 116 cm during years of 2000-2018. The deepest active layer appeared at QT05, which locates at the margin of permafrost from taliks formed by the thermal influences from the tributaries of Yangtze River headwaters, Tongtian river and Tuotuo river. Its average thickness was 307 cm from 2004 to 2013, where the surface vegetation is alpine meadow. In the continuous permafrost zone of QXP, which include Ch06, QT08, QT01, QT03, and Ch01 sites, the shallowest active layer located at the Kunlun Mountains pass (Ch06) under nearly bare land surface vegetation condition. Its average thickness was 147 cm during years of 2005-2018. The deepest active layer located at Wudaoliang (QT08) under bare land. Its average thickness was 235 cm during years of 2010-2018. For representative alpine meadow conditions, for example QT01 at Wudaoliang and Ch01 at Fenghuo Mountains, their average thicknesses were 163 cm and 167 cm. While at Beiluhe (QT03), about 10 km north of Ch01 site, its average thickness was about 231 cm with typical alpine meadow condition, which is apparently larger than QT01 and Ch01. In addition, the QT09 locates at the north limit permafrost distribution region, named Xidatan. Its average active layer thickness was 141 cm during years of 2011-2018 under typical alpine meadow condition. On the whole, in our opinion, the ground surface vegetation conditions may have some influences on active layer thickness spatial distribution. But it is not a control factors, especially at large spatial scale. The spatial distribution of active layer thickness was jointly influenced by climate conditions, ground temperature (including ground surface temperature and permafrost layer temperature), soil water content, soil texture. Due to the great spatial variation of these above influencing factors, the active layer thickness within our monitoring regions presented as great spatial variation.

In terms of time variation, all the monitoring sites showed the same pattern. Their active layer thicknesses were increasing gradually. But their increasing rate were very different amount sites,

with the largest increasing rate of 3.9 cm/yr at Ch01 and the lowest increasing rate of 0.8 cm/yr at
QT05. Of which worth noting is that the active layer thickness increasing rate is very sensitive to
the statistical period. Taking QT09 for example, its average increasing rate was 3.0 cm/yr during
2011-2018. While during years of 2014-2018, its average increasing rate was 6.9 cm/yr. So the
statistical active layer thickness increasing rates can't be considered as a long term thickness
increasing trend. It only revealed that the active layer thickness has a slow increase trend with inter-
annual fluctuation, and their increasing amplitudes are very different amount different monitoring
sites.

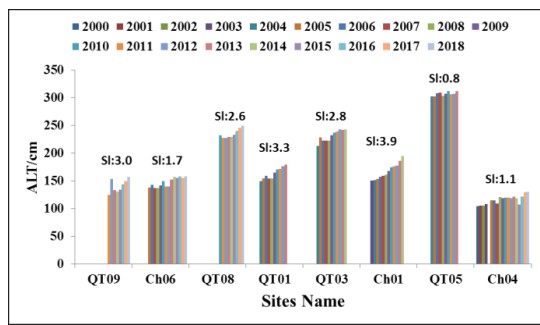


**Figure 7**. Variation in active layer thickness among different sites. SI represents the active layer thickness average
annual increasing rate.

## 3.3 Ground temperature

### 3.3.1 Temperature in the active layer

In this section, we choose ground temperature at 10cm depth and at the base of active layer of
years 2011-2013, during which all eight active layer monitoring sites had continuous ground
temperature monitoring data series, to analyze the active layer ground temperature spatial
distribution and their influence on active layer thickness spatial distribution (Table.5). The ground
temperature (ALT_Base_GT) was derived from geothermal interpolation when there was no
temperature probe at the real active layer depth position at the base of active layer. For all 8 active
layer monitoring sites, the mean annual ground temperature (10cm_GT) varied greatly from site to
site at 10cm depth. The lowest 10cm_GT appeared at Kunlun Mountains region (Ch06),which is -
2.86℃. For QT03, QT05 and Ch04, the 10cm_GT were positive, and as high as 1.12℃ and 1.25℃
at sites QT05 and Ch04. For ALT_Base_GT, the relative low temperature all appeared at mountain





regions, such as Ch06 at Kunlun Mountains and Ch01 at Fenghuo Mountains. This because the
ALT_Base_GT was simultaneously influenced by ground surface temperature and underlain
permafrost temperature, and in mountains regions, the permafrost layer temperature is often very
low in QXP. At the marginal regions of permafrost distribution or island permafrost region, such as
QT09, QT05 and Ch04, the ALT_Base_GT were relatively higher than other sites due to their high
underlain permafrost layer temperature.

**Table. 3** The mean active layer thickness, ground temperature at depth of 10 cm and permafrost table

| Sites Name | ALT/cm | 10cm_GT/°C | ALT_Base_GT/°C |
|------------|--------|------------|----------------|
| QT09 | 137 | -1.3 | -1.34 |
| Ch06 | 146 | -2.86 | -2.68 |
| QT08 | 228 | -1.64 | -1.45 |
| QT01 | 176 | -1 | -1.7 |
| QT03 | 241 | 0.03 | -1.29 |
| Ch01 | 180 | -1.35 | -2.47 |
| QT05 | 308 | 1.12 | -0.17 |
| Ch04 | 120 | 1.25 | -0.51 |

The scatter plot between active layer thickness and 10 cm_GT showed that, on the whole, ALT

increased with the increasing of 10 cm_GT, but they are not linear dependent (Fig.8). Especially for
Ch04 at island permafrost region under swamp meadow surface vegetation, the relationship between
ALT and 10 cm_GT was very different from other monitoring sites. This demonstrates that surface
ground temperature spatial distribution did have influence on ALT distribution, but it can't be used
as a main control factor for ALT prediction under different soil and vegetation conditions. Contrast
to the relationship between ALT and 10cm_GT, the relationship between ALT and ALT_Base_GT
is much better (Fig.3.2-2 b). If without considering the large deviation of sites QT09 and Ch04,
active layer thickness was nearly linear dependent on the variation of ALT_Base_GT, which
indirectly showed that the underlain permafrost temperature properties have great influence on ALT
distribution.

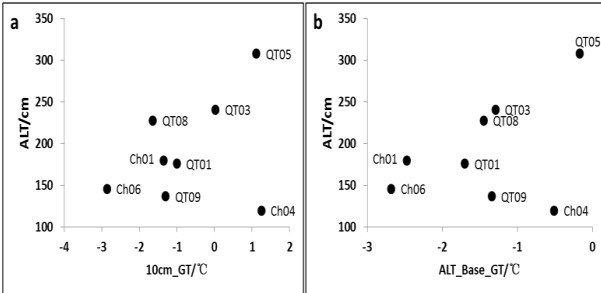


**Figure 8.** The relationship between active layer thickness and the temperature of permafrost table

In the dataset, the shallow ground temperature of 8 active-layer monitoring sites were collected

with automatic data logger along the Qinghai-Tibetan Road. At these sites, the annual mean ground
temperature at 10cm ranged from -2.62 ℃ to -0.20 ℃, while the mean temperature near top of
permafrost ranged from -2.69 ℃ to -0.37 ℃. The temperature at two depths has a good linear
correlation. The mean ground temperatures near the top of permafrost at 6 sites were 0.30 ℃ to
1.83 ℃ lower than the temperature of 10 cm. At only 2 sites (CN06 and QT08), the former is slightly
higher than the latter (approximately 0.2 ℃). The subsurface ground temperature of 10cm at all the
sites showed increasing trends with increase rates ranging from 0.03 to 0.19 ℃ per year, and the
maximum rate occurred at site QT09 which locates the northern marginal region of permafrost. The
increasing rate at the bottom of the active layer (near top of permafrost) is slightly lower than rate
of surface active layer. Even at CN06, there was a slight cooling trend at the bottom of the active
layer.
**3.3.2 Ground temperature from boreholes**

Fifteen borehole sites automatically collected ground temperature at different depths; 14 of

them located in the permafrost regions and only one is located in a structural talik region (QTB11).

Annual mean ground temperatures at depth of 3 m and 6 m are listed in Table L1. The ground

temperature of these two horizons at most sites not only has obvious seasonal variation, but also has
remarkable inter-annual variation. Except for QTB11 that locates in the seasonal frozen ground
region, the available mean annual ground temperatures at 10m and 20m are respectively shown in
the Figure 10. For the temperature of 10 m, the highest permafrost temperature appears at site
QTB05 that locates in Qumar River along the Qinghai-Tibetan Road, the mean annual ground
temperature of which is very close to 0 ℃, and meanwhile the active layer thickness has
approximately exceeded 9 m. The lowest temperature appears at site FCKGT that locates in high
plain area in the south of the Altun Mountain, where the permafrost temperature reaches -4 °C due
to extremely cold and dry climatic conditions. The ground temperature at all 15 boreholes showed
significant linear increasing trends, and the permafrost has warmed at different rates. The warming
rates at depth of 10m ranged from 0.02 °Cper decade (FCKGT) to 0.78 °C per decade (QTB05), but
increasing rates of temperature at depth of 20m were much smaller than 10 m, and varied between
0 °C per decade and 0.24 °C per decade. The annual mean temperature of 20 m at site ZNHGT has
rarely changed during the period from 2013 to 2018. At this depth, the most significant warming
occurred at site QTB02, QTB18 and QTB15. The warming rate of permafrost seems to have a strong
relationship with the temperature of permafrost itself.

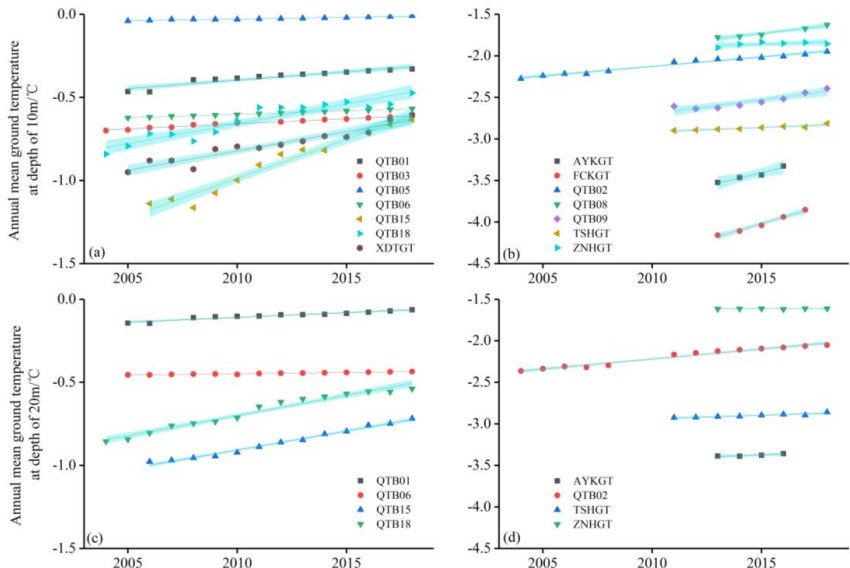


**Figure 9.** Annual mean ground temperature as a function of time at depth of 10m(a, b) and 20m(c, d) from

borehole with continuous data series

Figure 10a shows that the change rate of ground temperature at two shallow depths (10 cm and

the depth near top of permafrost). They show a trend of increasing first and then decreasing as the
temperature near the bottom of the active layer rises. Both colder and warmer sites have relatively
smaller variation rate of ground temperature. The sites with ground temperature between -2 °C and
-1 °C have the greatest ground warming rate. However, the deep ground temperature shows another
pattern and lower temperature permafrost tend to have a great warming rate (Figure 10b). It is
consistent with previous research results at Qinghai-Tibetan Plateau, and the correlation between
permafrost temperatures and warming rates is more significant. It indicates that the ice-water phase
transition effect in the conversion from permafrost to melting soil has significantly slowed the
process of ground temperature increase.
We also analyzed another 62 sites of which the ground temperatures are recorded manually.
The altitude of these sites ranges from 4142 to5247 m a.s.l. The drilling depth of borehole reached
10m at most of the sites, several reach 20m. The observation interval is once every one year or two
years., the multi-year averages based on single observations are calculated to compare the thermal
regime of different sites (Table L2). The multi-year mean ground temperature of 10m observed at
different sites ranged from -3.84 ℃ to 3.36 ℃. There are 10 observation fields with a positive mean
ground temperatures of 10 m and 52 fields with negative values. The site with highest ground
temperature is HT01, and the one with lowest temperature is STG. For all observation sites, the
ground temperature shows a slightly downward trend as the elevation increases.

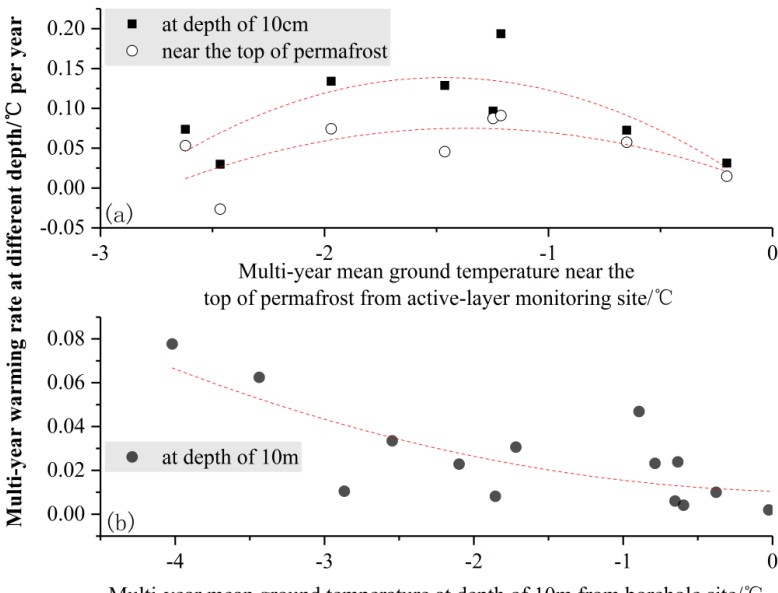


**Figure 10.** The relationship between warming rate and multi-year mean ground temperature during observation
period from active-layer monitoring site (a) and borehole site (b).



**3.3 Soil moisture**


The average volumetric soil water content (VWC) within ALT were calculated with depth-
weighted average method at time of ground surface begin to freeze and ALT reached its max
thawing depth at each monitoring sites (Fig.11a). In terms of inter-annual change, VWC had no
obvious changing trend with random inter-annual fluctuations. In terms of spatial variation, the
VWC varied from 0.141 to 0.403 $m^3/m^3$ among our monitoring sites, with the largest VWC at Ch04
and the lowest at QT08. Active layer soil water content was basically controlled by ground surface
vegetation conditions, soil texture and local drainage conditions. For example, it was swamp
meadow at Ch04 with about 60 cm depth of peat soil layer beneath ground surface. This resulted in
the very shallow active layer thickness and nearly saturated soil water content condition. At QT05,
the soil pit excavated in 2007 revealed that it was sand within 140 cm. This site has very bad
drainage condition and resulted in relatively high VWC, averaged 0.292 $m^3/m^3$ during 2004-2018.
While at QT08, where the soil type is also sand within active layer, because of its very good drainage
condition, VWC is very low, averaged 0.141 during 2012-2018.
Converting the VWC into total soil water depth per unit area that stored within active layer,
soil water depth varied from 290 mm to 890 mm among our monitoring sites (Fig.11b). QT05 had
the highest soil water depth, averaged 890 mm during 2004-2008. High soil water depth must absorb
high heat energy during active layer thawing process. This can explain why the active layer
thickness increasing rate was very low, while its ground surface temperature was very high.

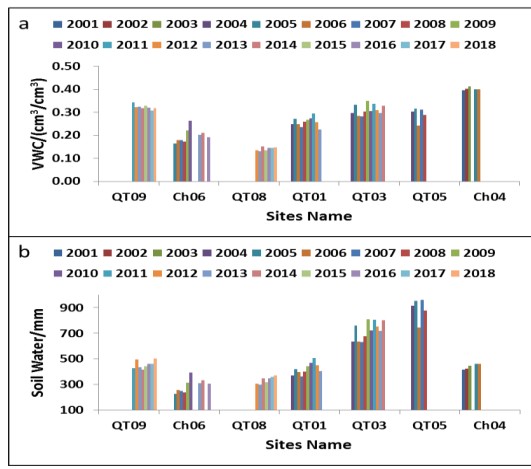


**Figure 11.** Variation in volumetric water content and soil water equivalent among different sites



## 4 Data availability

All datasets in this paper have been released and can be free download from the National Tibetan Plateau/Third Pole Environment Data Center (https://data.tpdc.ac.cn/en/disallow/789e838e-16ac-4539-bb7e-906217305a1d/,doi: 10.11888/Geocry.tpdc.271107) or Cryosphere Research Station on Qinghai-Xizang Plateau (http://new.crs.ac.cn/).

## 5 Conclusions

The observation data in permafrost regions on the QXP can provide basic data for the study of land-atmosphere interaction and climate change research. They could provide accurate inputs and verifications data for land surface models, reanalysis data and remote-sensing products, and climate models. The results revealed that the annual mean air temperatures of all 6 sites are between -2.3 ~ -6 ºC, and their seasonal variation characteristics are significant. Precipitation shows a significant seasonal change trend, which is closely related to the monsoon period. The annual mean air temperature of the XDT and TGL stations showed increasing trends, with rates of 0.66 and 0.40 ºC/10a, respectively, and ground temperature has significant warming trend. The precipitations show an insignificant week decrease trend. The active layer thickness has a slow increase trend with inter-annual fluctuation, and their increasing amplitudes are very different amount different monitoring sites. In addition, from the high-quality comprehensive dataset with a focus on permafrost thermal state on the QXP, which could provide accurate and effective forcing data and evaluation data for different models. This valuable permafrost dataset is worth maintaining and promoting in the future due to hard-won. It also provides a prototype of basic data collection and management for other permafrost regions.

**Author contributions.** L Zhao generated and designed the observation network, participated the field installation of most of the observations sites, found supports for maintaining the observation systems. DF Zou, GJ Hu, TH Wu, XD Wu, R Li, EJ Du, GY Liu, YP Qiao and X Yao participated the field works and maintained the observation sites. GJ Hu, R Li, EJ Du, GY Liu, X Yao and DF Zou performed data processing, organization and analyses. GJ Hu, L Zhao, EJ Du, GY Liu, X Yao and DF Zou wrote the paper, and all authors participated the manuscripts revision.

**Competing interests**. No conflict of interest.

**Acknowledgements.** We would like to thank all the scientists, engineers, and students who
participated in the field work and maintain this observation network and data acquisition.
**Financial support.** This work was financially supported by the National Natural Science
Foundation of China (41931180), the Second Tibetan Plateau Scientific Expedition and Research
(STEP) program, China (2019QZKK0201), and the National Natural Science Foundation of China

(42071094, 41701073).

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
