# Peer review of "A synthesis dataset of permafrost thermal state for"

_Earth System Science Data, 2021_

## Author Comment (AC2)

**Table S1** The location information of observation sites

| Observation items | Sites Name | Latitude ($°N$) | Longitude ($°E$) |
|---|---|---|---|
| Meteorological Stations | XDT | 35.72 | 94.13 |
| | TGL | 33.07 | 91.94 |
| | LDH | 31.82 | 91.74 |
| | ZNH | 35.49 | 91.96 |
| | AYK | 37.54 | 88.80 |
| | TSH | 35.62 | 94.06 |
| Active Layer Observation Stations | Ch01 | 34.73 | 92.89 |
| | Ch04 | 31.82 | 91.74 |
| | Ch06 | 35.62 | 94.06 |
| | QT01 | 35.14 | 93.04 |
| | QT03 | 34.82 | 92.92 |
| | QT05 | 33.96 | 92.34 |
| | QT08 | 35.22 | 93.09 |
| | QT09 | 35.72 | 94.13 |
| Boreholes Observation Stations | TSHGT | 35.36 | 79.55 |
| | FCKGT | 37.46 | 88.57 |
| | AYKGT | 37.52 | 88.61 |
| | ZNHGT | 35.49 | 91.96 |
| | QTB01 | 35.72 | 94.08 |
| | QTB02 | 35.63 | 94.06 |
| | QTB03 | 35.52 | 93.78 |
| | QTB05 | 35.36 | 93.45 |
| | QTB06 | 35.29 | 93.27 |
| | QTB08 | 35.22 | 93.08 |
| | QTB09 | 35.13 | 93.03 |
| | QTB15 | 33.10 | 91.90 |
| | QTB18 | 31.82 | 91.74 |
| | XDTGT | 35.72 | 94.13 |
| | QTB11 | 34.39 | 92.66 |
| | WQ01 | 35.36 | 99.13 |
| | WQ04 | 35.26 | 99.22 |
| | WQ07 | 35.39 | 99.30 |
| | WQ12 | 35.48 | 99.40 |
| | WQ19 | 35.48 | 99.50 |
| | ZK001 | 35.69 | 79.49 |
| | ZK002 | 35.73 | 79.46 |
| | ZK003 | 35.72 | 79.46 |
| | ZK004 | 35.79 | 79.42 |
| | ZK005 | 35.72 | 79.37 |
| | ZK006 | 35.76 | 79.38 |
| | ZK007 | 35.77 | 79.40 |

| | | |
|---|---|---|
| ZK009 | 35.86 | 79.39 |
| ZK016 | 34.54 | 80.42 |
| ZK017 | 34.62 | 80.64 |
| ZK018 | 34.62 | 80.62 |
| ZK020 | 34.59 | 80.32 |
| ZK022 | 34.57 | 80.40 |
| ZK024 | 34.63 | 80.39 |
| ZK025 | 34.64 | 80.39 |
| ZK026 | 34.56 | 80.39 |
| ZK027 | 34.57 | 80.39 |
| k308+860 | 36.43 | 77.58 |
| k514+950 | 35.84 | 79.40 |
| k520+050 | 35.80 | 79.41 |
| k529+100 | 35.73 | 79.45 |
| k572+000 | 35.40 | 79.55 |
| k582+000 | 35.32 | 79.54 |
| ZK036 | 33.05 | 84.16 |
| ZK044 | 33.18 | 85.31 |
| ZK045 | 33.21 | 85.35 |
| ZK046 | 33.39 | 85.63 |
| ZK048 | 33.39 | 85.63 |
| ZK049 | 33.39 | 85.63 |
| ZK050 | 33.35 | 85.65 |
| ZK052 | 33.80 | 85.13 |
| ZK053 | 33.39 | 85.63 |
| AYK02 | 37.52 | 88.61 |
| AYK03 | 37.51 | 88.70 |
| AYK04 | 37.54 | 88.79 |
| AYK05 | 37.54 | 88.83 |
| STG | 37.57 | 88.60 |
| STGK | 37.58 | 88.60 |
| FZB | 37.61 | 88.59 |
| HT02 | 37.66 | 88.68 |
| ZNH02 | 35.50 | 91.96 |
| ZNH03 | 35.50 | 91.96 |
| ZNH04 | 35.50 | 91.96 |
| ZNHX | 35.49 | 91.86 |
| KXL01 | 35.53 | 92.28 |
| KXL03 | 35.48 | 92.96 |
| KXL04 | 35.39 | 93.22 |
| HT01 | 37.75 | 88.72 |
| ZK008 | 35.86 | 79.37 |
| ZK012 | 35.80 | 79.03 |
| ZK013 | 35.80 | 79.03 |

| | | |
|---|---|---|
| ZK019 | 34.64 | 80.39 |
| ZK032 | 32.95 | 84.04 |
| ZK033 | 32.91 | 84.07 |
| ZK034 | 33.07 | 84.15 |
| ZK035 | 33.07 | 84.15 |
| ZK042 | 33.03 | 84.03 |
| ZK043 | 33.16 | 85.29 |
| ZK051 | 33.45 | 85.77 |
| AYK06 | 37.54 | 88.87 |
| WQ10 | 35.40 | 99.33 |
| ZK015 | 35.36 | 79.55 |
| ZNH01 | 35.50 | 91.96 |

---

## Author Response (AR1)

Title: A synthesis dataset of permafrost thermal state for the Qinghai-Tibet (Xizang) Plateau, China

**Dear Editor,**

Thank you very much for your great efforts dealing with the manuscript, and we appreciate the editors very much for their constructive comments and suggestions. We have replied the editor's comments carefully. The manuscript has been revised to the best with our knowledge according to the suggestions.

**Review1:**

The permafrost is a very important component on the QXP, and these datasets are also important for the permafrost community. The manuscript construction is well, although the English writing should be improved. My main concerns are about your datasets and their descriptions.

**General comment:**

1. Question

Line130: Table 1 is a summary on all observations:

(1) The number of active layer sites is 8 (in text) or 10 (in table)? I checked the dataset files which show 8 sites.

**Response:**

Thanks a lot. We have checked the dataset, and the active layer sites is 12. We have revised it in Table 1. **Line 96.**

**Table 1** The observation instruments and items for meteorological data, ground temperature and soil water content

| Observation site type | Available sites | Observation item | Instrument | Accuracy | Height/Depth | Frequencies |
|---|---|---|---|---|---|---|
| **Meteorological Stations** | 6 | Upward/downward short-wave radiation | CM3, Kipp & Zonen, Holland | ±10% | 2 m | 1/2 hour |
| | | Upward/downward long-wave radiation | CM3, Kipp & Zonen, Holland | ±10% | 2 m | |
| | | Air temperature | HMP45C, Vaisala Finland | ±0.5 ℃ | 2,5,10 m | |
| | | Air humidity | | ±3% RH | 2,5,10 m | |
| | | Wind velocity | 05103_L/RM, Campbell, USA | ±0.3 m/s | 2,5,10 m | |
| | | Precipitation | T-200B Precipitation Gauge | ±0.1 mm | 5 m away | |
| **Active Layer** | 12 | Soil temperature | 105T/109 Thermocouple temperature sensor | ±0.1 ℃ ±0.2 ℃ | 0.5 m,1.0 m,2 m, >2 m | 1/2 hour |
| | | Soil moisture content | CS616/ Hydra Soil moisture sensor | ±2.5% | | |
| **Borehole (automatic)** | 15 | Ground Temperature | Thermistor, SKLFSE, CHINA | ±0.05 ℃ | 3, 6, 10, 20 m | 1 hour |

| Observation site type | Available sites | Observation item | Instrument | Accuracy | Height/Depth | Frequencies |
|---|---|---|---|---|---|---|
| **Borehole (manual)** | 69 | Ground Temperature | Thermistor, SKLFSE, CHINA | ±0.05 ℃ | 10, 20 m | 1 year |

(2) The boreholes were measured in two kinds, automatic and manual, which should have different sampling time intervals. There are 77 boreholes (in text Line 97), but 40 sites (in table's second column). Authors should provide the numbers of these two kinds of measures correctly.

**Response:**

There are 84 boreholes, automatic and manual were 15 and 69. We have provided the numbers in Table 1. **Line 96.**

**Table 1** The observation instruments and items for meteorological data, ground temperature and soil water content

| Observation site type | Available sites | Observation item | Instrument | Accuracy | Height/Depth | Frequencies |
|---|---|---|---|---|---|---|
| **Meteorological Stations** | 6 | Upward/downward short-wave radiation | CM3, Kipp & Zonen, Holland | ±10% | 2 m | 1/2 hour |
| | | Upward/downward long-wave radiation | CM3, Kipp & Zonen, Holland | ±10% | 2 m | |
| | | Air temperature | HMP45C, Vaisala Finland | ±0.5 ℃ | 2,5,10 m | |
| | | Air humidity | | ±3% RH | 2,5,10 m | |
| | | Wind velocity | 05103_L/RM, Campbell, USA | ±0.3 m/s | 2,5,10 m | |
| | | Precipitation | T-200B Precipitation Gauge | ±0.1 mm | 5 m away | |
| **Active Layer** | 12 | Soil temperature | 105T/109 Thermocouple temperature sensor | ±0.1 ℃ ±0.2 ℃ | 0.5 m,1.0 m,2 m, >2 m | 1/2 hour |
| | | Soil moisture content | CS616/ Hydra Soil moisture sensor | ±2.5% | | |
| **Borehole (automatic)** | 15 | Ground Temperature | Thermistor, SKLFSE, CHINA | ±0.05 ℃ | 3, 6, 10, 20 m | 1 hour |
| **Borehole (manual)** | 69 | Ground Temperature | Thermistor, SKLFSE, CHINA | ±0.05 ℃ | 10, 20 m | 1 year |

(3) The observation frequencies should be provided in the table.

**Response:**

We have provided the observation frequencies in Table 1.

**Table 1** The observation instruments and items for meteorological data, ground temperature and soil water content

| Observation site type | Available sites | Observation item | Instrument | Accuracy | Height/Depth | Frequencies |
|---|---|---|---|---|---|---|
| **Meteorological Stations** | 6 | Upward/downward short-wave radiation | CM3, Kipp & Zonen, Holland | ±10% | 2 m | 1/2 hour |

| | | | | | |
|---|---|---|---|---|---|
| | | Upward/downward long-wave radiation | CM3, Kipp & Zonen, Holland | ±10% | 2 m |
| | | Air temperature | HMP45C, Vaisala Finland | ±0.5 ℃ | 2,5,10 m |
| | | Air humidity | | ±3% RH | 2,5,10 m |
| | | Wind velocity | 05103_L/RM, Campbell, USA | ±0.3 m/s | 2,5,10 m |
| | | Precipitation | T-200B Precipitation Gauge | ±0.1 mm | 5 m away |
| **Active Layer** | 12 | Soil temperature | 105T/109 Thermocouple temperature sensor | ±0.1 ℃ ±0.2 ℃ | 0.5 m,1.0 m,2 m, >2 m | 1/2 hour |
| | | Soil moisture content | CS616/ Hydra Soil moisture sensor | ±2.5% | | |
| **Borehole (automatic)** | 15 | Ground Temperature | Thermistor, SKLFSE, CHINA | ±0.05 ℃ | 3, 6, 10, 20 m | 1 hour |
| **Borehole (manual)** | 69 | Ground Temperature | Thermistor, SKLFSE, CHINA | ±0.05 ℃ | 10, 20 m | 1 year |

2. Question

Section 2 Monitoring data:

(1) You mentioned there are active layer (ground temperature and soil water content) and borehole observation in meteorological sites (Line 105-108). However, I did not find these data in the meteorological dataset file. Please mention it and keep the same name of site if these data were provided in the active layer data file and borehole data file.

**Response:**

Thanks for the review. We are sorry that due to the site naming rules, the names of different observation systems at the same site are not completely consistent. The latitude and longitude can be used to determine whether the stations are consistent. Furthermore, due to data integrity issues, not all active layers and borehole data of all weather stations are complete. We have provided the available data in the dataset, the site name of meteorological stations, active layer and borehole corresponds to following table:

Table The name of the same sites for meteorological stations, active layer and borehole

| Meteorological Stations | Active Layer | Borehole |
|---|---|---|
| XDT | QT09 | XDTGT |
| ZNH | ZHHAL | ZNHGT |
| AYK | AYKAL | AYKGT |
| TSH | TSHAL | TSHGT |
| TGL | QT04 | TGLGT |
| LDH | Ch04 | QTB18 |

We changed the related description to "The active layer observation system and GT borehole were set up simultaneously to record the permafrost, climate, vegetation, soil indices in different regions of the QTP.". **Line 106-108.**

(2) Figure 3 b and c, authors should provide the depth of each layer.

**Response:**

Thanks for the review. We have added the depth and changed Figure 3 b and c as follows:

[Figure]

**Figure 3.** The comprehensive observation system: (a) meteorological observation, (b) ground temperature and soil water content in the active layer and (c) ground temperature observation for permafrost.

It must be noted that the observed depth of active layer is different from site to site, and we only given the data with the same observation depth in the article.

(3) Line 147-152: What is the depth and layers for the ground temperature in the borehole. These data were provided in the borehole file? If yes, please separated into different sheets so that users can understand these data better.

**Response:**

We have clarified it to "…, which were downloaded to the depths of 3 m, 6 m, 10 m and 20 m depths within a steel pipe in the boreholes. All the borehole GTs along the QXH and located at the same sites with AMSs were measured at 15 minutes. The averaged value for each hour was automatically recorded by data loggers (CR1000/ CR3000, Campbell Scientific Company, Logan, UT, USA).". **Line 148-151.**

3. Question

Section 2.2 data processing workflow: There are three levels (raw data, daily, annual mean) in

 Anyway, it will be better if authors can provide the data in each lever.

**Response:**

The monthly and annual means air and ground temperatures were used to analysis from some sites, and our datasets hope to provide the raw daily data. We have revised it to "The monthly and annual mean air and GTs, radiation, wind speed, relative humidity and soil water content were also analyzed.". **Line 162-163.**

4. Question
Active layer dataset:
(1) I would suggest that Section 3.2, Section 3.3.1 and Section 3.3 soil moisture should be combined in one section. Section 3.2 active layer thickness should be read as Section 3.2 active layer data, which will be consistent with Table 1.

**Response:**

Thanks for the review. According to the comments, Section 3.2, Section 3.3.1 and Section 3.3 were reorganized as Section 3.2, including: Section 3.2.1 variation of active layer thickness, Section 3.2.2 Temperature in the active layer, Section 3.2.3 soil moisture in the active layer. Corresponding chart, table and text were corrected. **Line 220-318.**

5. Question
Boreholes dataset:
(1) 3.3.2 should be read 3.3 Ground temperature from boreholes.

**Response:**

Thanks. According to the comment, boreholes temperature part was organized as one section, Section 3.3 Permafrost temperature, Corresponding chart and text were corrected. **Line 319-367.**

(2) I checked the borehole data file and found many missing data, which should be mentioned in the data evaluation. How these missing data can influence on results in Figure 9?

**Response:**

Thanks. Most of them are not missing data in Figure 9 (c) and (d), because the observation sites in Figure 9 (c) and (d) have been established since 2010. All of them were located in the hinterland of the Plateau far away from roads or in no man's areas, and the data were collected almost annually once a year. However, due to the bad natural environment or the influence of nature reserve policy, some sites can observe by every two years or longer time. Therefore, it could cause some missing values. The data is very precious, we analyzed them to revealed the ground temperature variation trend in different permafrost region, and has given the confidence interval. In fact, it can be seen that the variation trends of ground temperature at 10 m or 20 m depths were relative stable, especially at 20 m depth, and the missing data can little affect on the warming trend.

**Review2:**

**General comments:**

The authors of the manuscript released a synthesis field dataset that include meteorological data at 6 stations, soil temperature and moisture in the active layer at 10 sites, and ground temperature measurement at 40 (or 77?) boreholes over the Qinghai-Tibet Plateau. The dataset is very valuable for geoscience community in Third pole. However, the readability of the manuscript needs to be greatly improved before it is accepted in ESSD.

**Response:**

Thanks. The permafrost monitoring network include 6 automatic meteorological stations, 12 active layer sites and, 84 boreholes. **Line 95-96.**

**Major comment:**

1.  Question

For data file, some basic information, such as geographical coordinate, landscape, soil type for each station, site or borehole, needs to be replenished. Active layer thickness data used in your analysis should be released also.

**Response:**

Thanks for the review. We have provided the geographical coordinate, landscape, soil type for each station, site or borehole in Table S1.

**Table S1** The location information of observation sites

| Observation items | Sites Name | Latitude (°N) | Longitude (°E) | Vegetation | Soil type |
|---|---|---|---|---|---|
| Meteorological Stations | XDT | 35.72 | 94.13 | Alpine meadow | Aridisols |
| | TGL | 33.07 | 91.94 | Alpine meadow | Gelisols |
| | LDH | 31.82 | 91.74 | Alpine wet meadow | Entisols |
| | ZNH | 35.49 | 91.96 | Alpine desert | Entisols |
| | AYK | 37.54 | 88.8 | Alpine desert | Inceptisols |
| | TSH | 35.36 | 79.55 | Alpine desert | Gelisols |
| Active Layer Observation Stations | Ch01 | 34.73 | 92.89 | Alpine meadow | Aridisols |
| | Ch04 (LDH) | 31.82 | 91.74 | Alpine wet meadow | Entisols |
| | Ch06 | 35.62 | 94.06 | Alpine steppe | Inceptisols |
| | QT01 | 35.14 | 93.04 | Alpine meadow | Gelisols |
| | QT03 | 34.82 | 92.92 | Alpine meadow | Gelisols |
| | QT05 | 33.96 | 92.34 | Alpine meadow | Gelisols |
| | QT08 | 35.22 | 93.08 | Alpine dessert | Aridisols |
| | QT09 (XDT) | 35.72 | 94.13 | Alpine meadow | Aridisols |
| | ZNH | 35.49 | 91.96 | Alpine desert | Entisols |
| | AYK | 37.54 | 88.8 | Alpine dessert | Inceptisols |
| | TSH | 35.36 | 79.55 | Alpine dessert | Gelisols |
| | QT04 (TGL) | 33.07 | 91.94 | Alpine meadow | Gelisols |
| | TSHGT | 35.36 | 79.55 | Alpine desert | Gelisols |

| | | | | | |
|---|---|---|---|---|---|
| | FCKGT | 37.46 | 88.57 | Alpine desert | Aridisols |
| | TGLGT | 33.07 | 91.94 | Alpine meadow | Gelisols |
| | AYKGT | 37.52 | 88.61 | Alpine desert steppe | Inceptisols |
| | ZNHGT | 35.49 | 91.96 | Alpine desert | Entisols |
| | QTB01 | 35.72 | 94.08 | Alpine steppe | Aridisols |
| | QTB02 | 35.63 | 94.06 | Alpine steppe | Aridisols |
| | QTB03 | 35.52 | 93.78 | Alpine desert steppe | Gelisols |
| | QTB05 | 35.36 | 93.45 | Alpine steppe | Gelisols |
| | QTB06 | 35.29 | 93.27 | Alpine steppe | Gelisols |
| | QTB08 | 35.22 | 93.08 | Alpine dessert | Aridisols |
| | QTB09 | 35.13 | 93.03 | Alpine meadow | Gelisols |
| | QTB15 | 33.1 | 91.9 | Alpine meadow | Aridisols |
| | QTB18 (LDH) | 31.82 | 91.74 | Alpine wet meadow | Entisols |
| | XDTGT | 35.72 | 94.13 | Alpine meadow | Aridisols |
| | QTB11 | 34.39 | 92.66 | Alpine meadow | Aridisols |
| | WQ01 | 35.36 | 99.13 | Alpine steppe | Inceptisols |
| | WQ04 | 35.26 | 99.22 | Alpine steppe | Inceptisols |
| | WQ07 | 35.39 | 99.3 | Alpine steppe | Inceptisols |
| | WQ12 | 35.48 | 99.4 | Alpine steppe | Aridisols |
| | WQ19 | 35.48 | 99.5 | Alpine wet meadow | Aridisols |
| Boreholes | ZK001 | 35.69 | 79.49 | Alpine desert steppe | Gelisols |
| Observation Stations | ZK002 | 35.73 | 79.46 | Alpine desert steppe | Gelisols |
| | ZK003 | 35.72 | 79.46 | Alpine desert steppe | Gelisols |
| | ZK004 | 35.79 | 79.42 | Alpine desert steppe | Gelisols |
| | ZK005 | 35.72 | 79.37 | Alpine desert steppe | Gelisols |
| | ZK006 | 35.76 | 79.38 | Alpine desert steppe | Gelisols |
| | ZK007 | 35.77 | 79.4 | Alpine steppe | Gelisols |
| | ZK009 | 35.86 | 79.39 | Alpine desert steppe | Gelisols |
| | ZK016 | 34.54 | 80.42 | Alpine steppe | Gelisols |
| | ZK017 | 34.62 | 80.64 | Alpine steppe | Inceptisols |
| | ZK018 | 34.62 | 80.62 | Alpine steppe | Entisols |
| | ZK020 | 34.59 | 80.32 | Alpine steppe | Entisols |
| | ZK022 | 34.57 | 80.4 | Alpine desert steppe | Gelisols |
| | ZK024 | 34.63 | 80.39 | Alpine steppe | Entisols |
| | ZK025 | 34.64 | 80.39 | Alpine steppe | Entisols |
| | ZK026 | 34.56 | 80.39 | Alpine steppe | Gelisols |
| | ZK027 | 34.57 | 80.39 | Alpine steppe | Gelisols |
| | k308+860 | 36.43 | 77.58 | Alpine desert | Entisols |
| | k514+950 | 35.84 | 79.4 | Alpine desert steppe | Gelisols |
| | k520+050 | 35.8 | 79.41 | Alpine desert steppe | Gelisols |
| | k529+100 | 35.73 | 79.45 | Alpine desert steppe | Gelisols |
| | k572+000 | 35.4 | 79.55 | Alpine desert | Gelisols |
| | k582+000 | 35.32 | 79.54 | Alpine desert steppe | Gelisols |

| | | | | |
|---|---|---|---|---|
| ZK036 | 33.05 | 84.16 | Alpine steppe | Entisols |
| ZK044 | 33.18 | 85.31 | Alpine meadow | Aridisols |
| ZK045 | 33.21 | 85.35 | Alpine steppe | Aridisols |
| ZK046 | 33.39 | 85.63 | Alpine steppe | Aridisols |
| ZK048 | 33.39 | 85.63 | Alpine wet meadow | Aridisols |
| ZK049 | 33.39 | 85.63 | Alpine steppe | Aridisols |
| ZK050 | 33.35 | 85.65 | Alpine steppe | Gelisols |
| ZK052 | 33.8 | 85.13 | Alpine steppe | Gelisols |
| ZK053 | 33.39 | 85.63 | Alpine steppe | Aridisols |
| AYK02 | 37.52 | 88.61 | Alpine desert steppe | Inceptisols |
| AYK03 | 37.51 | 88.7 | Alpine desert steppe | Inceptisols |
| AYK04 | 37.54 | 88.79 | Alpine desert steppe | Entisols |
| AYK05 | 37.54 | 88.83 | Alpine desert steppe | Inceptisols |
| STG | 37.57 | 88.6 | Alpine steppe | Gelisols |
| STGK | 37.58 | 88.6 | Alpine steppe | Gelisols |
| FZB | 37.61 | 88.59 | Alpine steppe | Gelisols |
| HT02 | 37.66 | 88.68 | Alpine steppe | Aridisols |
| ZNH01 | 35.5 | 91.96 | Alpine desert | Entisols |
| ZNH02 | 35.5 | 91.96 | Alpine desert steppe | Entisols |
| ZNH03 | 35.5 | 91.96 | Alpine desert steppe | Entisols |
| ZNH04 | 35.5 | 91.96 | Alpine desert steppe | Entisols |
| ZNHX | 35.49 | 91.86 | Alpine meadow | Inceptisols |
| KXL01 | 35.53 | 92.28 | Alpine wet meadow | Inceptisols |
| KXL03 | 35.48 | 92.96 | Alpine wet meadow | Inceptisols |
| KXL04 | 35.39 | 93.22 | Alpine wet meadow | Gelisols |
| HT01 | 37.75 | 88.72 | Alpine steppe | Entisols |
| ZK008 | 35.86 | 79.37 | Alpine steppe | Aridisols |
| ZK012 | 35.8 | 79.03 | Alpine desert | Aridisols |
| ZK013 | 35.8 | 79.03 | Alpine desert | Aridisols |
| ZK019 | 34.64 | 80.39 | Alpine steppe | Entisols |
| ZK032 | 32.95 | 84.04 | Alpine steppe | Inceptisols |
| ZK033 | 32.91 | 84.07 | Alpine steppe | Entisols |
| ZK034 | 33.07 | 84.15 | Alpine steppe | Entisols |
| ZK035 | 33.07 | 84.15 | Alpine steppe | Entisols |
| ZK042 | 33.03 | 84.03 | Alpine meadow | Inceptisols |
| ZK043 | 33.16 | 85.29 | Alpine meadow | Inceptisols |
| ZK051 | 33.45 | 85.77 | Alpine steppe | Aridisols |
| AYK06 | 37.54 | 88.87 | Alpine steppe | Inceptisols |
| WQ10 | 35.4 | 99.33 | Alpine steppe | Inceptisols |
| ZK015 | 35.36 | 79.55 | Alpine desert | Gelisols |

We have released the active layer thickness data in Table 3.

**Table. 3** The mean active layer thickness, ground temperature at depth of 10 cm and permafrost table

| Sites Name | ALT/cm | 10cm_GT/°C | ALT_Base_GT/°C |
|---|---|---|---|

| | | | |
|---|---|---|---|
| QT09 | 137 | -1.3 | -1.34 |
| Ch06 | 146 | -2.86 | -2.68 |
| QT08 | 228 | -1.64 | -1.45 |
| QT01 | 176 | -1 | -1.7 |
| QT03 | 241 | 0.03 | -1.29 |
| Ch01 | 180 | -1.35 | -2.47 |
| QT05 | 308 | 1.12 | -0.17 |
| Ch04 | 120 | 1.25 | -0.51 |

2. Question

Name of station, active layer site, and key place name used in text should be shown in Figure 1. The permafrost types you mentioned in text should be indicated in the figure also. Figure 3 is not informative or redundant.

**Response:**

Thanks for the review. Figure 1 has redrawn and added the name of station, active layer site, and key place name. Based on the continuity, the permafrost can be classified to four types: continuous (90%), discontinuous (50-90%), sporadic (10-50%), and isolated (0-10%) permafrost (Brown et al., 1997).

In this study, the permafrost type of site was defined by field survey and expert knowledge. The regional distribution of permafrost was from the reference of Zou et al. (2017), however the permafrost continuity has not compiled yet. So, only the permafrost type information of site was described in the text.

[Figure]

**Figure 1.** The permafrost monitoring networks on the QXP. AL: active layer; AWS: automatic meteorological

station

We have added the depth and changed Figure 3 b and c as follows:

[Figure]

**Figure 3.** The comprehensive observation system**:** (a) meteorological observation, (b) ground temperature and soil water content in the active layer and (c) ground temperature observation for permafrost.

3. Question

English needs to be greatly improved.

**Response:**

Thanks. We have checked and revised the English.

**Specific comments:**

1. Question

Line 20: Qinghai-Tibet Plateau is formal and should be used to replace the Qinghai-Xizang (Tibet) Plateau.

**Response:**

We have revised "Qinghai-Xizang (Tibet) Plateau" to "Qinghai-Tibet Plateau (QTP)". **Line 19-20.**

2. Question

Line 81: in very high, high what, elevation?

**Response:**

Here is elevation, and we have revised it to "…and its environmental factors in high-elevation and cold-climate regions of the QTP." **Line 81-82.**

3. Question
   Line 97: 40 boreholes in table 1 but here is 77

**Response:**
   The boreholes were 84. We have checked and revised it throughout the revised version.

4. Question
   Line 102: this sentence can be moved to introduction section.

**Response:**
   Thanks. We have moved it. **Line 67-69.**

5. Question
   Line 108: What is LDH? Give the full name for the first time?

**Response:**
   It was Liangdaohe (LDH) site. We have given the full name. **Line 108.**

6. Question
   Figure 2, the abbreviation of stations name in the text should consistent with the figure.

**Response:**
   Thanks. We have revised it as follow:

[Figure]

**Figure 2.** The six comprehensive meteorological stations

7. Question

Section 2.3: The quality control process, including sensor calibration, may need to be supplemented.

**Response:**

We have clarified it to "The instruments at meteorological stations are calibrated every few years by comparing observations with standard instruments for about one week." **Line 159-160.**

8. Question

Line 157: Here you're mentioned ALT but data file is missing.

**Response:**

Here we only describe how the active layer thickness can be obtained from soil temperature observations, and the ALT data was provided in Table 3.

**Table. 3** The mean active layer thickness, ground temperature at depth of 10 cm and permafrost table

| Sites Name | ALT/cm | 10cm_GT/°C | ALT_Base_GT/°C |
|---|---|---|---|
| QT09 | 137 | -1.3 | -1.34 |
| Ch06 | 146 | -2.86 | -2.68 |
| QT08 | 228 | -1.64 | -1.45 |
| QT01 | 176 | -1 | -1.7 |
| QT03 | 241 | 0.03 | -1.29 |
| Ch01 | 180 | -1.35 | -2.47 |
| QT05 | 308 | 1.12 | -0.17 |
| Ch04 | 120 | 1.25 | -0.51 |

9. Question

Line 210: What meaning of the significance here?

**Response:**

The sentence ", and has a good significance" has no meaning here and has been deleted. **Line 211.**

10. Question

Line 228: The deepest active layer located at Wudaoliang and the deepest active layer appeared at QT05 in line 222 are very confusing, please clear it. Is it meaningful to distinguish continuous permafrost and sporadic island permafrost here?

**Response:**

Thanks. At QT05, the average thickness was 307cm, which was apparently deeper than QT08 (235cm). The main reason for this phenomenon is that ground surface temperature (10 cm depth) at QT05 was very high, about 1.16 ℃. While at QT08, the ground surface temperature was -1 ℃. It can't be considered a meaningful sign of sporadic island permafrost distribution at QT05. In fact, there is large area of permafrost distribution at this site.

11. Question
    Figure 7, a line plot may be better.

**Response:**

Thanks. The line plot for Figure7 was shown as follow, but it was not better than the original figure for showing the interannual variation of active layer thicknesses at different sites. So the Figure 7 was not changed.

[Figure]

12. Question
    Line 281: Where is the Fig.3.2-2b?

**Response:**

This is a text error. It was Fig.8b, and was corrected in the revision. **Line 282.**

13. Question
    Line 299: the section title is confusing with section 3.3.1, using "permafrost temperature"?

**Response:**

Thanks. We have revised it. **Line 319.**

14. Question
    Line 302: Where is the Table L1?

**Response:**

It was ground temperature dataset, and we have deleted it.

15. Question
Line 318: More discussions are needed to clear this found.

**Response:**

The next paragraph is actually a further discussion of this found. In the revised draft, we put this sentence at the beginning of the next paragraph for better logic. **Line 339-351.**

**Response:**

It has been modified and figure 9 has been cited in the text. **Line 331.**

17. Question
Line 324 to 325: Why? This needed to more discussion.

**Response:**

Thanks. For now, this is only an observed phenomenon because of the lack of adequate sites. Judging from the location and topography of existing sites, regional climate and local topography may be the main reasons for this phenomenon. More detailed discussion was added to the revised text.

The warming rate of permafrost seems to have a strong relationship with the temperature of permafrost itself. Fig. 11a shows that the change rate of GT at two shallow depths (10 cm and the depth near top of permafrost). They show an increasing trend first and then decreasing as the temperature near the bottom of the active layer rises. Both colder and warmer sites have a relatively lower variation rate of GT. The sites with GTs between -2 ℃ and -1 ℃ have the greatest ground warming rate. The warming of the active layer in permafrost regions may be mainly related to regional climate and local topography. Because most sites (QT1,QT3,QT8) with the largest warming rates are located on the high plain in the interior of the QTP, and they are geographically relatively close to each other. The two sites (CN1,CN6) with the lowest GT are located in the mountain areas (respectively belong to Fenghuo Mountain and Kunlun Mountain). At the same time, the other two sites (CN4, QT5) with the highest GT are located in the regions with the warmest climatic conditions, although the underlying surfaces are substantially different. Further study is necessary because the current number of sites is far from enough. **Line 348-351.**

**Response:**

It was ground temperature dataset, and we have deleted it.

Line 340: A figure and more discussions are needed to clear the elevation-dependent warming of ground temperature.

**Response:**

Thanks. In the original manuscript, this sentence refers to the change of ground temperature with elevation, rather than the relationship between warming rate of permafrost and elevation.

20. Question
Line 358: How do you do this conversion?

**Response:**

The total soil water depth was calculated through soil water content (VWC) multiply active layer thickness (ALT).

We would like to express our great appreciation to you for comments on our paper.
Looking forward to hearing from you.
Thank you and best regards.

Yours sincerely,
Lin Zhao

Corresponding author:
Name: Lin Zhao
E-mail: lzhao@nuist.edu.cn
Name: Guojie Hu
E-mail: huguojie123@lzb.ac.cn

---

## Author Response (AR2)

Title: A synthesis dataset of permafrost thermal state for the Qinghai-Tibet (Xizang) Plateau, China

**Dear Editor,**

Thank you very much for your great efforts dealing with the manuscript, and we appreciate the editors very much for their constructive comments and suggestions. We have replied the editor's comments carefully. The manuscript has been revised to the best with our knowledge according to the suggestions.

**Comments to the Author:**

I am pleased to inform you that we got several very positive comments on your manuscript (ms: essd-2021-1), although the editor and I have a similar concern. Therefore, I would like to ask you modify the manuscript before it can be acceptable for publication.

**The main concern is about figures and tables.**

1. Question

One cannot read the information from Table 2 because there are so many values. Do you think it can be a part of datasets? The main context provides the necessary information of these six stations only.

**Response:**

Thanks a lot. We have checked the dataset, and the information in Table 2 was removed duplicate data in the dataset. We have revised Table 2 as follows:

**Table 2.** The information of six meteorological stations

| Sites | XDT | TGL | LDH | ZNH | AYK | TSH |
|---|---|---|---|---|---|---|
| Elevation (m a.s.l) | 4538 | 5100 | 4808 | 4784 | 4300 | 4844 |
| Ta (°C) | -3.6 | -4.7 | -2.3 | -4.9 | -5.2 | -6.0 |
| RH (%) | 53.5 | 51.5 | 48.2 | 53.9 | 46.1 | 40.6 |
| Precipitation (mm) | 384.5 | 352.0 | 388.6 | 277.8 | 158.6 | 103.3 |
| Wind speed (m/s) | 4.1 | 4.1 | 3.2 | 4.7 | 4.5 | |
| DSR ($W/m^2$) | 224.2 | 233.4 | 231.4 | 204.8 | 198.2 | 250.8 |
| USR ($W/m^2$) | 66.8 | 61.4 | 46.6 | 46.3 | 53.8 | 68.5 |
| DLR ($W/m^2$) | 223.0 | 214.8 | 237.2 | 233.8 | 223.0 | 211.5 |
| ULR ($W/m^2$) | 304.5 | 304.5 | 315.9 | 303.2 | 307.6 | 311.3 |
| Net radiation | 75.9 | 82.3 | 106.0 | 89.2 | 59.8 | 82.5 |

2. Question

Figure 5, 6, 7, and 9 are barely readable. Especially, Figure 7&9 are both not very intuitive and

I don't think anyone can discern the individual years. This can be collapsed or a presented differently?

**Response:**

Thanks a lot. Figure 7&9 has revised as follows:

[Figure]

**Figure 7**. Variation in active layer thickness among different sites. SI represents the active layer thickness average

annual increasing rate.

[Figure]

**Figure 9.** Variation in volumetric water content and soil water equivalent among different sites

**Others:**

1. Question

   Table1 & Line145: The accuracy of ground temperature is 0.05 in Table, while 0.1 in Line 145.

**Response:**

   Thanks a lot. We have revised it to 0.1 °C in Table1.

2. Question

   L170: You mentioned the seasonal variations of thermal variables on the QTP, I would like to suggest the four seasons should be defined due to the longer winter and the shorter summer in the special region.

**Response:**

   Thanks a lot. We have defined it to" The seasonal (spring (Mar.–May), summer (Jun.–Aug.), autumn (Sept.–Nov.), and winter (Dec.–Feb.)) variation of air temperature at all 6 sites is significant with the annual mean from -2.3 to -6 °C (Fig. 5).". **Line 170-171.**

3. Question

   Figure 9: Not the CH01 site?

**Response:**

   Thanks. Here is the CH04 site.

4. Question
    L369: free should be read freely.

**Response:**

Thanks. We have revised it to "All datasets in this paper have been released and can be free download from the National Tibetan Plateau/Third Pole Environment Data Center (https://data.tpdc.ac.cn/en/disallow/789e838e-16ac-4539-bb7e906217305a1d/, doi: 10.11888/ Geocry.tpdc.271107), and more information about the Permafrost Monitoring Network on the Qinghai-Tibet Plateau can be found at Cryosphere Research Station on Qinghai-Xizang Plateau (http://new.crs.ac.cn/). ". **Line 373-374.**

The last one, but the most important, is that the manuscript is a data paper. The theme should relate to measures, quality control, uncertainties, potential applications, and the data analysis should be reduced greatly. Please revised the whole of manuscript based on this principle.

**Response:**

Thanks. We have described in detail to the data of the measurement, quality control, uncertainties in section 2:

2 Monitoring networks and data processing
    2.1 Permafrost monitoring networks
    2.2 Monitoring data
    2.3 Data processing workflow

And, the potential applications were introduced in the conclusions section as follows:

"They could provide accurate inputs and verifications for land surface models, reanalysis data and remote-sensing products, and climate models." …. "In addition, the high-quality comprehensive dataset with a focus on permafrost thermal state on the QTP could provide accurate and effective forcing data and evaluation data for different models. This valuable permafrost dataset is worth maintaining and promoting in the future due to hard-won. It also provides a prototype of basic data collection and management for other permafrost regions." **Line 376-390.**

The data analysis section is mainly a simple analysis and introduction of the hydro-thermal characteristics of permafrost. This section is to show the quality of the data, and to demonstrate the basic characteristics of the permafrost observations data, which could let the reader know what these data can do and how they can be used.

We would like to express our great appreciation to you for comments on our paper.
Looking forward to hearing from you.
Thank you and best regards.

Yours sincerely,
Lin Zhao

Corresponding author:
Name: Lin Zhao

E-mail: lzhao@nuist.edu.cn
Name: Guojie Hu
E-mail: huguojie123@lzb.ac.cn